# Simple and Fast Prediction of Gestational Diabetes Mellitus Based on Machine Learning and Near-Infrared Spectra of Serum: A Proof of Concept Study at Different Stages of Pregnancy

**DOI:** 10.3390/biomedicines12061142

**Published:** 2024-05-21

**Authors:** Daniela Mennickent, Lucas Romero-Albornoz, Sebastián Gutiérrez-Vega, Claudio Aguayo, Federico Marini, Enrique Guzmán-Gutiérrez, Juan Araya

**Affiliations:** 1Departamento de Ciencias Básicas y Morfología, Facultad de Medicina, Universidad Católica de la Santísima Concepción, 4090541 Concepción, Chile; dmennickent@ucsc.cl; 2Departamento de Análisis Instrumental, Facultad de Farmacia, Universidad de Concepción, 4070386 Concepción, Chile; luromero@udec.cl; 3Departamento de Bioquímica Clínica e Inmunología, Facultad de Farmacia, Universidad de Concepción, 4070386 Concepción, Chile; tm.sebastiangutierrez@gmail.com (S.G.-V.); caguayo@udec.cl (C.A.); 4Department of Chemistry, University of Rome La Sapienza, 00185 Rome, Italy; federico.marini@uniroma1.it

**Keywords:** gestational diabetes mellitus, first trimester, second trimester, near-infrared spectroscopy, serum samples, predictive models, machine learning

## Abstract

Gestational diabetes mellitus (GDM) is a hyperglycemic state that is typically diagnosed by an oral glucose tolerance test (OGTT), which is unpleasant, time-consuming, has low reproducibility, and results are tardy. The machine learning (ML) predictive models that have been proposed to improve GDM diagnosis are usually based on instrumental methods that take hours to produce a result. Near-infrared (NIR) spectroscopy is a simple, fast, and low-cost analytical technique that has never been assessed for the prediction of GDM. This study aims to develop ML predictive models for GDM based on NIR spectroscopy, and to evaluate their potential as early detection or alternative screening tools according to their predictive power and duration of analysis. Serum samples from the first trimester (before GDM diagnosis) and the second trimester (at the time of GDM diagnosis) of pregnancy were analyzed by NIR spectroscopy. Four spectral ranges were considered, and 80 mathematical pretreatments were tested for each. NIR data-based models were built with single- and multi-block ML techniques. Every model was subjected to double cross-validation. The best models for first and second trimester achieved areas under the receiver operating characteristic curve of 0.5768 ± 0.0635 and 0.8836 ± 0.0259, respectively. This is the first study reporting NIR-spectroscopy-based methods for the prediction of GDM. The developed methods allow for prediction of GDM from 10 µL of serum in only 32 min. They are simple, fast, and have a great potential for application in clinical practice, especially as alternative screening tools to the OGTT for GDM diagnosis.

## 1. Introduction

Gestational diabetes mellitus (GDM) is a hyperglycemic state of variable severity that is first diagnosed during pregnancy [1], with negative short- and long-term consequences on both maternal and fetal health [2]. Its worldwide prevalence is 14.7%, according to the International Association of Diabetes and Pregnancy Study Groups (IADPSG) criteria [3], very similar to what is reported in the Chilean population [4]. The diagnosis of this disease is typically made by an oral glucose tolerance test (OGTT) in the second or third trimester of pregnancy [5]. The OGTT is unpleasant [6,7], time-consuming [8,9], and has low reproducibility [10,11]. Moreover, by the time of its use, the fetal phenotype is already altered in GDM pregnancies [12,13,14]. Therefore, the diagnosis of GDM can be improved. Machine learning (ML) predictive modeling of biomedical relevant data is a powerful means of meeting that goal [15,16], either as an early detection tool, or as an alternative screening tool for OGTT.

Numerous models have been proposed to predict GDM at different stages of pregnancy. In 2022, our research group published a literature review to summarize the methodologies, results, and limitations of the latest ML-based work for the prediction of GDM. This review revealed that many of the predictive models for this pregnancy disease derive from data acquired by instrumental techniques, such as liquid or gas chromatography coupled to mass spectrometry (LC-MS or GC-MS, respectively), nuclear magnetic resonance (NMR) spectroscopy, and polymerase chain reaction (PCR), among others [17]. More recently, new articles have been published in the context of GDM diagnosis based on the same techniques, i.e., LC-MS [18,19,20,21], GC-MS [22], NMR spectroscopy [23], and PCR [24,25,26,27]. Methods based on these techniques are very time consuming, as they require tedious sample preparation procedures or prolonged instrumental runs. Consequently, simpler and faster strategies should be developed.

Near-infrared (NIR) spectroscopy is an analytical technique that is based on the absorption, emission, scattering, reflection, or diffuse reflection of light in the NIR range of the electromagnetic spectrum, i.e., between 12,500 and 4000 cm^−1^ [28]. Biomolecules are capable of interacting with NIR radiation and, therefore, an NIR spectrum constitutes the biochemical fingerprint of a biological sample [29]. NIR spectroscopy has multiple advantages, some of which are typical of vibrational spectroscopy, e.g., it is noninvasive, nondestructive, reagent-free, waste-free, simple, fast, low-cost, and requires minimal sample preparation [30]. Moreover, NIR spectroscopy is more versatile and less expensive than other vibrational spectroscopy techniques [28]. Due to its advantageous analytical features, this technique has been widely applied in different fields of science, including clinical diagnostics [31,32]. 

NIR spectroscopy has never been used as a diagnostic support tool for GDM. Therefore, its capability for GDM prediction at particular stages of pregnancy, such as before or at the time of GDM diagnosis, remains unexplored. This study aims to develop ML predictive models for GDM based on NIR spectroscopy, and to evaluate their potential as early detection or alternative screening tools according to their predictive power and time of analysis.

The contributions of this research are as follows:It tests for the first time ML models based on NIR spectroscopy as diagnostic support tools for GDM.It develops and evaluates a novel, simple, and rapid bioanalytical method for early detection and alternative screening of GDM, which avoids some of the disadvantages of OGTT, such as its unpleasant and time-consuming nature.It proposes an ML model based on NIR spectra of serum, which has similar or better predictive power than its literature counterparts, but with a shorter time of analysis, which makes it very attractive for use as an alternative screening tool to OGTT.It exhibits the potential of this new technology in obstetrics and gynecology, for example, for the prediction of other diseases and complications of pregnancy.

## 2. Materials and Methods

The workflow used to meet the objective of this study consisted of the following: subject recruitment, collection of medical data and serum samples, acquisition of NIR spectral data, and training and internal validation of the ML models.

### 2.1. Ethical Aspects

This work was approved by the Ethics Committee of Servicio de Salud Concepción (17-12-88) and was carried out in accordance with the Declaration of Helsinki. 

### 2.2. Subjects Recruitment

First trimester pregnant women were recruited at three primary health centers in Concepción, Chile: CESFAM Victor Manuel Fernández, CESFAM Santa Sabina, and CESFAM Tucapel. Recruitment was conducted between 2017 and 2019. Both primiparous and multiparous participants were included. Individuals with pregestational diabetes or any pregnancy alterations different than GDM were excluded. Subjects who gave their written informed consent were included in the study and followed up until the second trimester of pregnancy. Ultimately, 96 pregnant women participated in this work.

### 2.3. Medical Data Collection

A total of 28 medical variables were retrieved from CESFAM online records and subject’s self-reported statements. The former comprised age and body mass index (BMI) in the first trimester of gestation. The latter encompassed first trimester information, i.e., supplement consumption, hyperemesis, and vaginal bleeding, and preconception information, i.e., drug use, prior pregnancy diseases or complications, prior pregnancy non-viability, fertility issues, history of polycystic ovary syndrome (PCOS), age at menarche, the month of last period, personal morbid history, and family morbid history. 

### 2.4. Blood Sample Collection

Blood samples were collected in the first and the second trimester of pregnancy, after fasting (12 h) or after a 75 g glucose load (2 h). First and second trimester samples were taken before and at the time of GDM diagnosis, respectively. They were transported to laboratory at 4 °C. Sera and NaF/citrate plasma were obtained by centrifugation (10,000× *g*, 5 min, 4 °C). They were aliquoted and stored at −80 °C. 

### 2.5. NIR Spectra Acquisition

Sera were randomized before analyses. Each sample was thawed at room temperature, homogenized, and 10 µL were deposited and dried (37 °C, 30 min) on a MirrIR low-e reflective microscopic slide (Kevley Technologies, Chesterland, OH, USA). NIR spectra (range 10,500–4000 cm^−1^, resolution 4 cm^−1^) were acquired in transflectance mode using a FT-IR Spectrum Frontier/Spotlight 400 Microscopy System (Perkin Elmer, Waltham, MA, USA). The acquisition time was 2 min per spectrum. Five NIR spectra, i.e., instrumental replicates, were recorded and averaged per sample. 

### 2.6. GDM Diagnosis, Cohorts, and Study Groups

In the second trimester of pregnancy, pregnant women were subjected to an OGTT. This is the reference method to diagnose GDM in Chile. Fasting and post-load plasma glucose were quantified by the hexokinase method [33]. The Chilean diagnostic criteria were used, i.e., subjects with fasting glycemia between 100 and 125 mg/Dl, or post-load glycemia higher than 140 mg/Dl (75 g, 2 h), were diagnosed with GDM [34]. In this study, two cohorts were considered. Those cohorts were defined according to the availability of first and second trimester serum samples. Of the 96 participants in the study, 49 had only first trimester samples, 14 had only second trimester samples, and 33 had samples from both trimesters. The first cohort of this study had first trimester serum samples, from which NIR spectra were obtained. This cohort, from now on called the first trimester cohort, consisted of 82 pregnant women: 15 with GDM and 67 with normal glucose tolerance (NGT) (medical and NIR data are displayed in Appendix A, respectively). The second cohort of this study had second trimester serum samples, and second trimester NIR spectra were obtained. This cohort, from now on called the second trimester cohort, consisted of 47 subjects: 8 with GDM and 39 with NGT (medical and NIR data are presented in Appendix A, respectively). 

### 2.7. Classical Statistics Analyses

Qualitative medical data were compared by two-sided Fisher exact test. The normality of quantitative medical data was evaluated by Shapiro–Wilk test. Normally distributed parameters were compared using unpaired Student *t* test. Non-normally distributed parameters were compared using Mann–Whitney test. *P* values less than 0.05 were considered statistically significant. These analyses were carried out using GraphPad Prism version 9.5.1 (GraphPad Software Inc, Boston, MA, USA).

### 2.8. ML Analyses

#### 2.8.1. Data Pretreatment

Prior to ML analyses, qualitative medical parameters were transformed into categorical variables. NIR spectra were also transformed from reflectance to absorbance. In addition to the full NIR spectral range, three shorter wavenumber regions were also analyzed separately: 10,500–7600 cm^−1^, 7600–5100 cm^−1^, and 5100–4000 cm^−1^. For each spectral range, 80 different combinations of mathematical transformations were tested, including Savitzky–Golay smoothing or first/second derivative with varying filter width, standard normal variate scattering correction, weighted least squares baseline correction, and 2-norm normalization. The order in which these transformations were applied was based on recent specialized literature [35,36]. Medical and NIR data were preprocessed by autoscaling and mean centering, respectively.

#### 2.8.2. Single- and Multi-Block Analyses

For single-block analyses, pretreated data were analyzed by partial least squares linear discriminant analysis (PLS-LDA). PLS-LDA was chosen due to its ability to deal with a large number of highly collinear predictors, therefore allowing us to overcome the limitations connected to the use of linear discriminant analysis (LDA) on this type of data. This was accomplished by formulating the classification problem in terms of regression, so that the partial least squares (PLS) algorithm could be used for calculating the solution. PLS allow for the calculation of multivariate regression models in the presence of an ill-conditioned predictor matrix *X*. The *X* matrix is compressed into a set of scores *T*, having maximum covariance, with the response *y* to be predicted through a weight matrix *R*:(1)T=XR

The response is then regressed on the scores, according to the following: (2)$$\hat{y}=Tq\;$$
where $\hat{y}$ is the vector collecting the predicted responses and the coefficients q are called the y-loadings. By combining Equations (1) and (2), it can be shown that the regression model can be expressed in terms of the original variables: (3)$$\hat{y}=XRq=Xb\;$$

The regression coefficients b being given by Rq. To use a regression model for classification, it is necessary to use a binary-coded response, where the value 1 corresponds to GDM and 0 to NGT. PLS regression is then used to relate the binary-coded *y* to the spectral data *X*, as summarized by Equations (1)–(3). However, since the predicted response $\hat{y}$ is real-valued, it is necessary to define a threshold value *y_thres_*, so that if the predicted response is higher than the threshold then the individual is predicted as GDM, whereas if it is lower, they are predicted as NGT. In the present study, the threshold was calculated by applying LDA to $\hat{y}$. 

For multi-block analyses, pretreated data were analyzed by sequential and orthogonalized PLS-LDA (SO-PLS-LDA). SO-PLS-LDA is a generalization of PLS-LDA to the multi-block case, which relies on the use of sequential and orthogonalized partial least squares (SO-PLS) as a multi-block regression model to approximate the binary-coded response. SO-PLS, as the name suggests, involves the sequential calculation of PLS models between each predictor block and the response. Moreover, each block is orthogonalized with respect to the scores of the previous PLS regressions, so that it is possible to evaluate whether the addition of a new block brings a relevant improvement to the model or not. For the simplest case of two predictor blocks *X*_1_ and *X*_2_, the SO-PLS-LDA algorithm can be summarized as follows:
Calculate a PLS model between the binary-coded y and the first predictor block $X_{1}$: $\hat{y}=T_{1}q_{1}=X_{1}b_{1}$.Orthogonalize the second block $X_{2}$ with respect to $T_{1}$: $X_{2,\;orth}=X_{2}-T_{1}{\left(T_{1}^{T}T_{1}\right)}^{-1}T_{1}^{T}X_{2}$.Calculate a PLS model between the residuals of the first regression $e_{1}=y-\hat{y}$ and the orthogonalized second predictor block $X_{2,\;orth}$: ${\hat{e}}_{1}=T_{2,\;orth}q_{2,\;orth}=X_{2,\;orth}b_{2,\;orth}$.The overall model can then be written as: ${\hat{y}}_{SO}=X_{1}b_{1}+X_{2,\;orth}b_{2,\;orth}$, where ${\hat{y}}_{SO}$ collects the final predictions of the SO-PLS model. The classification model is obtained by applying LDA on ${\hat{y}}_{SO}$.

Further details on these classification ML techniques can be found elsewhere [37]. For multi-block analyses, different block orders were tested. Every model was subjected to double cross-validation (DCV), an intensive and robust internal validation strategy consisting of two nested cross-validation loops. The inner loop is used for model training and optimization, and the outer loop for model validation [38]. For DCV, the following parameters were used: 10 segments for the inner loop, 20 segments for the outer loop, and 50 repetitions. Models were developed using in-house written functions in MATLAB version R2021a (The MathWorks Inc, Natick, MA, USA). 

#### 2.8.3. Evaluation of Predictive Performance

Models’ predictive performance was evaluated by means of their specificity (Sp), sensitivity (Se), and non-error rate (NER). For the best models, the area under the receiver operating characteristic curve (AUROC) was also determined. The mathematical definition of these parameters can be found elsewhere [39,40]. In general terms, the specificity and the sensitivity denote the ability to correctly classify NGT and GDM subjects, respectively. The NER reflects the ability to correctly classify both NGT and GDM subjects, and the AUROC represents the overall predictive performance of the model in a graphical manner. These parameters were determined with respect to the reference method, for which figures of merit were assumed to be maximum. Every value is presented as the average ± the standard deviation of 50 repetitions in DCV.

#### 2.8.4. Variable Importance and Selection

For each model, variable importance in projection (VIP) scores were obtained. Variables with average VIP scores larger than 1 were considered as relevant for model performance [38]. This information was used for variable selection in the multi-block models, and for biochemical interpretation in the final models.

## 3. Results

The results of this work are presented in order of cohorts (first and second trimester), showing statistical description of medical variables, NIR spectra of sera, and predictive performance of the best ML models obtained.

### 3.1. First Trimester Cohort

#### 3.1.1. Description of the First Trimester Cohort

To characterize this cohort, classical statistical techniques were used. Table 1 displays 28 medical variables and compares their behavior in NGT and GDM pregnancies. In this cohort, the prevalence of GDM was 18.3%. Only two parameters are statistically different between the two groups: history of GDM in a prior pregnancy, and family history of diabetes mellitus (DM). Both are more frequent in the GDM group than in the NGT group. 

#### 3.1.2. Prediction of GDM with First Trimester Serum NIR Spectral Data

To predict GDM using the biochemical information that serum samples contain, NIR spectra were acquired. Figure 1 shows NIR spectra from first trimester NGT and GDM sera. The spectral traces’ behavior depends on the wavenumber range. In particular, signal sequential noise varies with wavenumber. There is a high-noise region between 10,500 and 7600 cm^−1^, a varying-noise region between 7600 and 5100 cm^−1^, and a low-noise region between 5100 and 4000 cm^−1^. Since different spectral ranges present different noise characteristics, they may require different mathematical pretreatments before ML analyses. Therefore, NIR spectra were divided in three ranges, according to their sequential noise features: Range 1, from 10,500 to 7600 cm^−1^ (R1); Range 2, from 7600 to 5100 cm^−1^ (R2), and Range 3, from 5100 to 4000 cm^−1^ (R3). Posterior analyses considered the three spectral regions, as well as the full range, from 10,500 to 4000 cm^−1^ (Full). 

First trimester NIR spectral data were used to develop different single-block predictive models for GDM. For every spectral region, Full, R1, R2, and R3, 80 combinations of pretreatments were tested (Appendix A). Table 2 presents the characteristics of the best models, i.e., the ones with the highest NER in DCV, for each spectral range. The NIR region with the best predictive performance is R1, with an NER of 0.6321 ± 0.0489. This value is moderately higher than that obtained with the Full spectral range of 0.5726 ± 0.0410. 

To assess if the latter models could be improved, NIR Full and NIR R1 data were combined with the 28 medical variables mentioned in Section 2.3. Different multi-block models were trained and validated (Appendix A). The addition of medical data does not improve the overall predictive performance of the original models, whether compared to the models based only in NIR spectra, or compared to models based only in medical data (NER of 0.6133 ± 0.0298 in DCV for a model based on the 28 medical variables, NER of 0.6592 ± 0.0000 in DCV for a model based on history of GDM in a prior pregnancy, and NER of 0.6692 ± 0.0000 in DCV for a model based on family history of DM, with history of GDM in a prior pregnancy and family history of DM being the statistically significant variables in Table 1). None of the multi-block models outperform the best single-block model obtained with NIR R1 data only. The simplification of the multi-block models through variable selection did not improve its predictive performance either. As in single-block analyses, NIR R1-based multi-block models tend to show moderately higher NERs than NIR Full-based multi-block models.

Figure 2 presents the overall predictive performance of the best ML model obtained using NIR spectra from first trimester serum samples. It corresponds to the NIR R1 spectral range (10,500–7600 cm^−1^) with pretreatment by normalization and mean centering. It predicts GDM with a DCV AUROC of 0.5768 ± 0.0635. The most relevant spectral intervals for the performance of this model, i.e., those mainly composed of variables with VIP scores larger than 1, are 10,500–9828 cm^−1^ and 8826–7858 cm^−1^. The tentative biomolecular assignment of these spectral intervals is shown in Appendix A.

### 3.2. Second Trimester Cohort

#### 3.2.1. Description of the Second Trimester Cohort

To characterize this cohort, classical statistical analyses were performed. Table 3 presents the same 28 medical parameters considered for the first trimester cohort and compares their behavior in NGT and GDM subjects. In this cohort, the prevalence of GDM was 17.0%. There are only two variables that statistically differ between the two groups: BMI and history of GDM in a prior pregnancy. The former is higher, and the latter is more frequent in GDM pregnancies than in NGT pregnancies.

#### 3.2.2. Prediction of GDM with Second Trimester Serum NIR Spectral Data

To predict GDM using the biochemical information contained in sera, NIR spectra were recorded. Figure 3 exhibits NIR spectra from second trimester NGT and GDM serum samples. Due to their sequential noise behavior, NIR spectra were divided into the same three regions considered for the first trimester cohort: R1 (10,500–7600 cm^−1^), R2 (7600–5100 cm^−1^), and R3 (5100–4000 cm^−1^). Subsequent analyses considered both the three NIR regions, and the Full spectral range (10,500 to 4000 cm^−1^).

Second trimester NIR spectral data were used to train and validate different single-block models for GDM prediction. For each spectral range (Full, R1, R2, and R3), 80 combinations of mathematical pretreatments were assessed (Appendix A). Table 4 displays the figures of merit of the best models, i.e., the ones with the highest NER in DCV, for every NIR region. The range with the greatest predictive power is R3, with an NER of 0.7894 ± 0.0431. This performance is much better than that obtained with the Full NIR range of 0.4642 ± 0.0321. 

To evaluate if the performance of the predictive models for this cohort could be enhanced, NIR Full and NIR R3 were combined with the 28 medical parameters mentioned in Section 2.3. Different multi-block models were developed (Appendix A). The combination of NIR Full with medical data improves the overall performance in comparison to the Full NIR single-block model. Likewise, the combination of NIR R3 with medical data increases the overall predictive power in comparison to models based on medical data only (NER of 0.6115 ± 0.0467 in DCV for a model based on the 28 medical variables, NER of 0.6642 ± 0.0159 in DCV for a model based on BMI, and NER of 0.6875 ± 0.0000 in DCV for a model based on history of GDM in a prior pregnancy, with BMI and history of GDM in a prior pregnancy being the statistically significant variables in Table 3). Nevertheless, the addition of medical data does not improve the predictive performance compared to the best single-block model, obtained with NIR R3 data only. The simplification of the multi-block models by means of variable selection does not outperform its predictive performance either. Similarly, with what was observed in single block-analyses, NIR R3-based multi-block models tend to present higher NERs than NIR Full-based multi-block models.

Figure 4 shows the overall predictive performance of the best ML model obtained by employing NIR spectra from second trimester sera. It corresponds to the NIR R3 spectral region (5100–4000 cm^−1^) with pretreatment by first derivative (width = 15) and mean centering. It predicts GDM with an AUROC of 0.8836 ± 0.0259 in DCV. The most relevant spectral intervals for the performance of this model, i.e., those mainly composed of variables with VIP scores larger than one, are 5028–4856 cm^−1^, 4764–4702 cm^−1^, 4492–4442 cm^−1^, 4392–4364 cm^−1^, 4302–4268 cm^−1^, 4206–4176 cm^−1^, and 4096–4000 cm^−1^. The tentative biomolecular assignment of these spectral intervals is presented in Appendix A.

## 4. Discussion

This work shows that ML modeling with NIR spectra from first trimester sera leads to a moderate performance for the prediction of GDM. It also shows that modeling with NIR data from second trimester samples results in a high predictive power for GDM. In both cases, the entire method takes only 32 min, considering both sample preparation and data acquisition by NIR spectroscopy. These findings suggest that the second trimester NIR-spectroscopy-based method could be used as an alternative screening tool for GDM.

### 4.1. The Addition of Medical Data Does Not Improve the Predictive Performance of NIR Data-Based Models 

The prevalence of GDM in the study cohorts was higher than what was reported for the Chilean population in 2015, this being 13.0% [4]. This behavior is consistent with the fact that the prevalence of GDM is increasing both in Chile [4] and worldwide [41]. The medical variables that differed between GDM and NGT groups were history of GDM in a prior pregnancy, family history of DM, and BMI. Higher frequencies or levels of these variables were observed in pregnant women with GDM. This observation makes sense since they are known risk factors for GDM, both in the Chilean [4] and global population [1]. The statistical behavior of these risk factors was not exactly the same in the first and the second trimester cohorts, however, in a multivariate scale the two cohorts behaved similarly. Full medical data allowed us to predict GDM with a very similar overall performance in the first and the second trimester cohorts. Moreover, the addition of medical data with multi-block techniques did not improve the performance of NIR data-based predictive models in any of the cohorts, even when all statistically relevant variables were part of the best multi-block models. 

Even though multi-block analysis is often associated with an increased predictive power compared to single-block analysis, this is not always the case [42]. One strategy that can improve predictive performance is to apply variable selection [38,40,43], however, in this study it did not have that effect. Another strategy is to use multi-block techniques of a higher data fusion level [42]. Multi-block analysis can be related to low-, mid-, or high-level data fusion (LLDF, MLDF, and HLDF, respectively). LLDF techniques work directly on the original data blocks; MLDF techniques operate on features extracted from each data block; and HLDF techniques fuse the outcome of models built from each data block [44]. The multi-block technique applied here, SO-PLS-LDA, corresponds to MLDF, since information is sequentially extracted from the different blocks to construct the model [37]. Therefore, the application of an HLDF technique may increase the predictive power of the models presented here. The use of such a technique should be considered carefully, since HLDF modeling is more complex, time-consuming, and more difficult to interpret [44].

### 4.2. NIR Data-Based Prediction Has Advantages over Medical Data Prediction 

NIR data-based models performed as well as or better than medical data-based models. In fact, the best model built on NIR spectral data from first trimester serum samples showed an overall performance similar to that obtained with full medical data in the same cohort. Likewise, the best model built on NIR spectral data from second trimester serum samples presented a much higher predictive power than the one obtained with full medical data in the same cohort. The same behavior was observed when comparing the best models based on NIR data with models based on the individual medical variables that showed statistical significance in each cohort: for the first trimester cohort, history of GDM in a prior pregnancy and family history of DM; and for the second trimester cohort, BMI and history of GDM in a prior pregnancy.

It is important to mention that all the medical parameters considered here are clinical. Clinical variables involve anthropometrical measurements, demographical parameters, and personal or family morbid history data. In general, models involving this type of variable are associated with a moderate performance for the prediction of GDM [17]. Furthermore, this kind of information is generally obtained by means of self-report questionnaires and, therefore, is subject to bias. In contrast, NIR spectral data are obtained through the objective instrumental analysis of biological samples. Hence, in addition to presenting a similar or higher predictive power than medical-data-based models, NIR data-based models are less subjective. 

### 4.3. NIR Spectral Data Pretreatment Is Essential to Maximize Predictive Power

In the two study cohorts, particular spectral regions achieved a better predictive performance than full spectral ranges. In the case of first trimester serum samples, R1 presented an NER moderately higher than that obtained with the Full range. In the case of second trimester serum samples, R3 exhibited an NER much higher than the one obtained with the Full range. This tendency was maintained when medical data were added. 

It is likely that spectral segmentation allowed us to better optimize data pretreatment. Pretreatment operations are used to remove chemically irrelevant sources of variation in the data, reducing the contribution of signals that are not related to the property being predicted, and improving the performance of both qualitative and quantitative analyses [35]. Typical pretreatment for infrared (IR) data includes selecting the optimal wavenumber range and correcting different spectral alterations, such as random and systematic noise, light scattering, and baseline shift, among others [35,36]. The effect of pretreatment on predictive power is highly data-dependent [36]. Therefore, it is important to optimize it depending on the spectral features that need to be corrected in each particular case. Here, signal sequential noise varied between NIR spectral regions, suggesting that they may require different pretreatments. Indeed, the optimal pretreatment for each spectral region was different, both between short spectral regions and compared to the Full NIR range. 

### 4.4. Predictive Performance in the First and the Second Trimester Is Related to Biochemical Changes Occurring throughout GDM

The optimal NIR ranges differed between trimesters. For first and second trimester sera, the spectral ranges with higher predictive power were 10,500–7600 cm^−1^ and 5100–4000 cm^−1^, respectively. This difference in optimal spectral range and associated predictive performance might be related to the biochemical changes that underlie the development of GDM.

The biochemical interpretation of NIR spectra is a challenging task, since the bands observed in NIR spectra are mainly due to overtones and combination bands of fundamental vibrational modes [28]. However, it is possible to tentatively relate spectral patterns to particular biomolecules [29,45,46]. In the first trimester best model, two NIR intervals stood out, 10,500–9828 cm^−1^ and 8826–7858 cm^−1^, whereas in the second trimester case, six spectral intervals did so, 5028–4856 cm^−1^, 4764–4702 cm^−1^, 4492–4442 cm^−1^, 4392–4364 cm^−1^, 4302–4268 cm^−1^, 4206–4176 cm^−1^, and 4096–4000 cm^−1^. These first and second trimester spectral intervals involve vibrations of various chemical bonds, among which there are some that have been associated with carbohydrates, lipids, and proteins. Therefore, these three biomolecules would be altered in GDM, in both trimesters of pregnancy.

Based on the tentative assignments made, the potential biochemical differences between the two trimesters are not evident. However, there is a key difference between them in GDM. The hyperglycemia state that characterizes GDM manifests only in the late second trimester or in the early third trimester of pregnancy [1]. In other words, while glycemia is not altered in the first trimester, it is altered in the second trimester. Interestingly, the optimal spectral range for predicting GDM in the second trimester (5100–4000 cm^−1^) has been identified as relevant for quantifying glucose in serum samples. Indeed, Goodarzi and Saeys showed that 2100–2300 nm (4762–4348 cm^−1^) was the most important NIR region for glucose quantification in human serum. They discussed that this was consistent with previous studies, which had identified 2000–2500 nm (5000–4000 cm^−1^) as the most informative wavelength range for glucose measurement [47]. Their result is coherent with the tentative assignments performed here, in which wavenumbers near 4762–4348 cm^−1^ were related to carbohydrates. In consequence, it is very likely that the best second trimester model achieves a better predictive performance than the best first trimester counterpart because it accounts for biochemical changes that become evident only when GDM is fully established.

### 4.5. NIR Data-Based Prediction Has Advantages over Other Instrumental Data-Based Prediction

The NIR-based models presented here allowed us to predict GDM in 32 min, considering sample preparation and spectral acquisition of each instrumental replicate. This is fast compared to other instrumental methods reported in literature. In the following paragraphs, the proposed method is compared with those presented in some other existing studies, both in terms of predictive power and duration of analysis. This comparison is summarized in Table 5. It is important to note the difficulty of directly comparing this work with others. Other studies that address the same problem using other instrumental techniques are extremely heterogeneous in terms of reporting their predictive power, with some reporting the value in the training phase, others in the internal validation phase, and others in the external validation phase. In addition, different articles use different types of internal and external validation strategies or different metrics to assess predictive power. On the other hand, most of these studies do not report the time required for each step of the analytical methods they propose, therefore it is only possible to have an estimated value of their duration of analysis. As a consequence, the following comparison should be read with caution.

There are studies applying LC-MS- or GC-MS-based methods to predict GDM at different stages of pregnancy. Their predictive performance varies, e.g., with AUROCs of 0.7075 [48], 0.745–0.797 [50], 0.724–0.902 [21], 0.729–0.906 [49], and 0.771–0.907 [22] before GDM diagnosis, and 0.7800 [48], 0.745–0.828 [50], and 0.83–0.90 [57] at the time of GDM diagnosis. Even though some of these methods achieve a high predictive power, they are very time-consuming. For instance, the LC-MS metabolomics strategy of Zhang et al. [48] takes approximately 1.5 h, with a sample preparation step of at least 15 min, and two LC-MS runs of 30 min each. Likewise, the LC-MS proteomics approaches of Guo et al. [49] and Wang et al. [21] include sample preparation processes that takes more than 4 h and 8 h, respectively. The GC-MS metabolomics methods of Raczkowska et al. [50] and Dudzik et al. [57] require a sample preparation procedure of more than 16 h. Similarly, the GC-MS metabolomics strategy of Zhu et al. [22] requires at least 1.5 h for sample preparation only, as indicated in the reference they cited in the methods section of their study [59].

Other articles use NMR spectroscopy metabolomics-based methods to predict GDM at different points in pregnancy [23,52,53,54]. Their predictive power is variable but tends to be better at the time of GDM diagnosis than earlier, e.g., with AUROCs of 0.62 and 0.59, respectively, in the study of McBride et al. [52], and NERs of 0.695–0.885 and 0.635–0.825, respectively, in the study of Pinto et al. [53]. Studies that focus particularly on early detection of GDM achieve moderate predictive power, with AUROCs of 0.610–0.719 [54] and 0.796 [23]. Papers presenting this kind of strategy do not usually mention details about duration of analysis in the methodology section. However, sample preparation of biological fluids for this type of analysis usually takes 1–1.5 h, while NMR data acquisition typically takes 4–5 min per sample [51].

There are also works employing PCR-based methods for GDM prediction. Some authors have based their methods on single nucleotide polymorphisms. Such methods are valid to predict GDM at any point of life. Their predictive power is moderate, e.g., with an NER of 0.531–0.552 [55] and an AUROC of 0.7694 [27]. Moreover, they are associated with long analysis times. For example, the strategy of Yu et al. [55] consists of the extraction of genomic DNA, and its analysis by PCR-restriction fragment length polymorphism (PCR-RFLP). The PCR-RFLP protocol alone takes more than 2 h. Likewise, the method of Zulueta et al. [27] consists of genomic DNA extraction, and its analysis by iPLEX-PCR using the MassARRAY system from Agena Bioscience. According to the manufacturer, the entire workflow for iPLEX MassARRAY PCR takes 8 h [60]. Some other authors have based their GDM predictive methods on micro-RNAs. Their predictive performance varies, e.g., with AUROCs of 0.600–0.669 [56] before GDM diagnosis, and AUROCs of 0.74–0.92 [58] at the time of GDM diagnosis. Although some of these methods reach a high predictive power, they involve long durations of analysis. For instance, the approaches of Zhao et al. [56] and Cao et al. [58] consist of RNA extraction, reverse transcription, and a TaqMan-based quantitative PCR (TaqMan-qPCR) that takes, alone, about 1 h. Furthermore, sample preparation for TaqMan-qPCR usually takes more than 1.5 h [61].

Even though the best first trimester method presented here is simple and fast, it showed a moderate performance for the prediction of GDM compared to that reported in literature with more time-consuming methods. This NIR data-based method could be improved by modifying sample preparation, e.g., by removing from sera the high concentration proteins that might be interfering with the analysis of lower concentration biomolecules, which could be important to differentiate the two study groups. This would increase the time of analysis, however, it could be adjusted, for example, by reducing the drying time. The simplicity and rapidity of this method, coupled with an improved predictive power, would make it ideal for the early detection of GDM. On the other hand, the best second trimester method presented here exhibited a very high predictive power, similar to or better than that reported in literature with much slower methods. This predictive performance, together with its simplicity and rapidity, makes it an excellent alternative screening method for GDM.

### 4.6. The Presented Strategy Has Advantages over Other IR-Based Strategies 

Before this work, NIR spectroscopy had never been assessed as a diagnostic support tool for GDM. There is only one study applying IR spectroscopy for the prediction of this pregnancy disease, that of Bernardes-Oliveira et al. The following paragraph compares the proposed method with that developed by them.

The method of Bernardes-Oliveira et al. consists of the analysis of plasma samples with attenuated total reflection Fourier transform mid-IR spectroscopy, and is able to predict GDM with an accuracy of 100% [62]. The limitation of this strategy is that plasma samples were collected in a very wide time range, at 9–39 or 12–38 weeks of pregnancy for the control or the GDM group, respectively. Therefore, even though their study showed the great potential of IR spectroscopy to differentiate subjects with and without GDM, it was not designed to predict GDM at particular stages of pregnancy. In contrast, in the present study serum samples were collected in the first and the second trimester of pregnancy, allowing us to evaluate the capability of IR spectroscopy to predict GDM both before and at the time of diagnosis.

### 4.7. The Proposed NIR Data-Based Method Has Advantages over the OGTT

The OGTT is the reference method for diagnosis of GDM in Chile [34] and in several other countries of the world [5]. However, as discussed in the introduction of this study, the OGTT is unpleasant, time-consuming, has low reproducibility, and results are tardy.

Even though none of the presented NIR spectroscopy-based models reach the diagnostic specificity and sensitivity of the OGTT, they have important advantages over it: they do not require pregnant women to go through the unpleasant process of taking a high glucose load, nor do they require subjects to spend two hours at the health center in order to obtain a blood sample. Moreover, they allow analysts to obtain a result from 10 µL of serum in only 32 min, which would ease the work of laboratory staff.

The method developed from NIR spectra of first trimester sera has the additional advantage of allowing the early detection of GDM, however, its predictive power is not high enough to be implemented in clinical reality. In contrast, the method developed from NIR spectra of second trimester sera has a high predictive power. As discussed above, the performance of this method is similar to or better than other instrumental methods reported in literature, which are much more complex and time-consuming. It is for these reasons that the second trimester NIR-based method is an attractive option for use as an alternative screening method to the OGTT. 

### 4.8. Strengths of This Study

To our knowledge, this is the first study reporting NIR spectra-based methods for the prediction of GDM, either as early detection or alternative screening tools. They are simpler and faster than other strategies proposed in literature when predicting this pregnancy disease. Moreover, the best second trimester model achieved a highly competitive predictive power compared to methods from literature, making it ideal as an alternative screening tool for GDM. In addition, NIR data pretreatment was performed in an exhaustive and systematic manner, enabling maximization of the predictive power for both first and second trimester sera. Finally, every model was subjected to DCV, allowing us to obtain reliable results despite a limited sample size. 

### 4.9. Limitations of This Study

The sample size is small. Models constructed on a limited number of samples are prone to overfitting. Even though DCV was used to minimize this effect, future external validation studies are needed to confirm the effectiveness of the developed methods to predict GDM in different populations. In addition, the predictive models for GDM presented here are restricted to the Chilean diagnostic criteria. Further studies should be performed to evaluate the performance of NIR spectra-based methods for the prediction of GDM under other diagnostic criteria. Finally, this is neither a longitudinal nor a paired study, so the comparison between first and second trimester results should be made with caution. 

## 5. Conclusions

In this work, NIR spectroscopy of serum samples was evaluated for the prediction of GDM at different stages of pregnancy. NIR data-based predictive models were methodically optimized and robustly validated. The developed methods are simple, fast, and have great potential for application as clinical decision support tools in medical practice. Even though the first trimester approach should still be improved for application as an early detection tool for GDM, the second trimester strategy presents characteristics that make it suitable to be used as an alternative screening tool to the OGTT at the time of GDM diagnosis, e.g., a high predictive power for GDM, simplicity, and rapidity. Further studies are needed to confirm these findings in other populations and under different GDM diagnostic criteria.

## Figures and Tables

**Figure 1 biomedicines-12-01142-f001:**
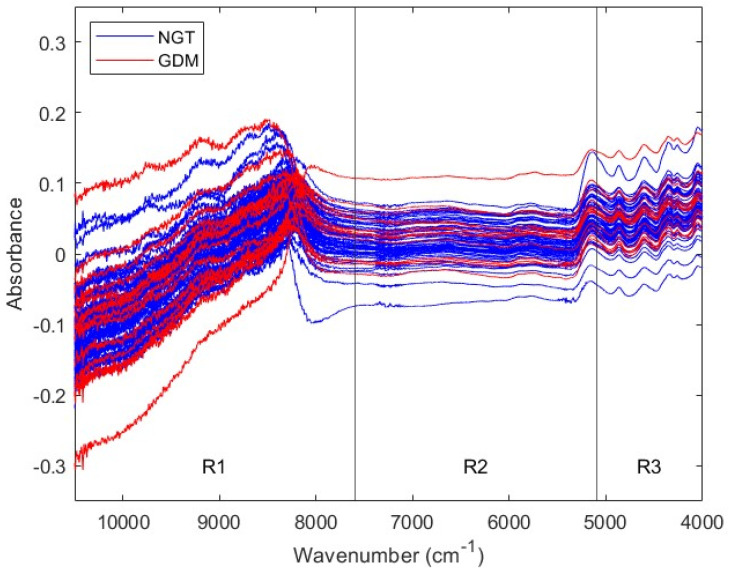
NIR spectra from first trimester serum samples. Each trace corresponds to the NIR spectrum of one serum sample, that is, of one subject. Spectra from NGT and GDM pregnant women are colored in blue and red, respectively. Four spectral ranges are considered: full, from 10,500 to 4000 cm^−1^; Range 1, from 10,500 to 7600 cm^−1^; Range 2, from 7600 to 5100 cm^−1^; and Range 3, from 5100 to 4000 cm^−1^. NIR: near-infrared; NGT: normal glucose tolerance; GDM: gestational diabetes mellitus; R1: Range 1; R2: Range 2; R3: Range 3.

**Figure 2 biomedicines-12-01142-f002:**
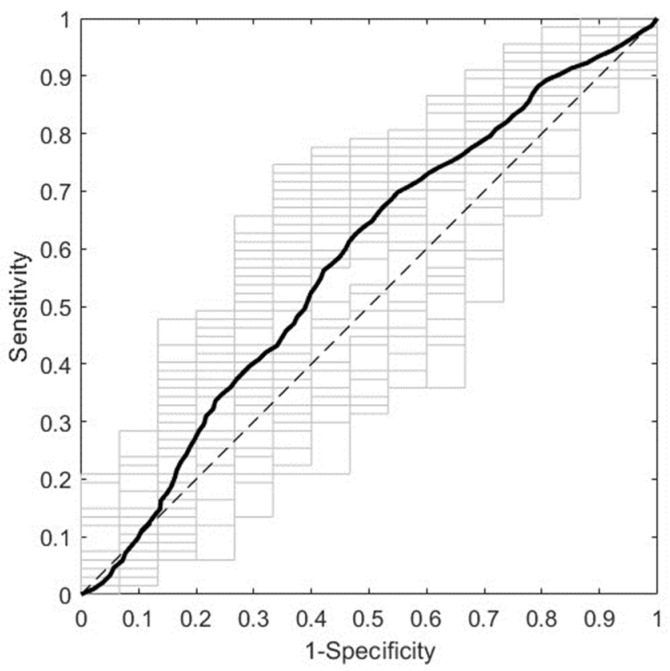
ROC curve of the best predictive model for the first trimester cohort. The model was trained with NIR spectra (R1, 10,500–7600 cm^−1^) from first trimester serum samples after pretreatment by normalization and mean centering. The average and the individual curves of 50 DCV repetitions are colored in black and gray, respectively. ROC: receiver operating characteristic; NIR: near-infrared; R1: Range 1; DCV: double cross-validation.

**Figure 3 biomedicines-12-01142-f003:**
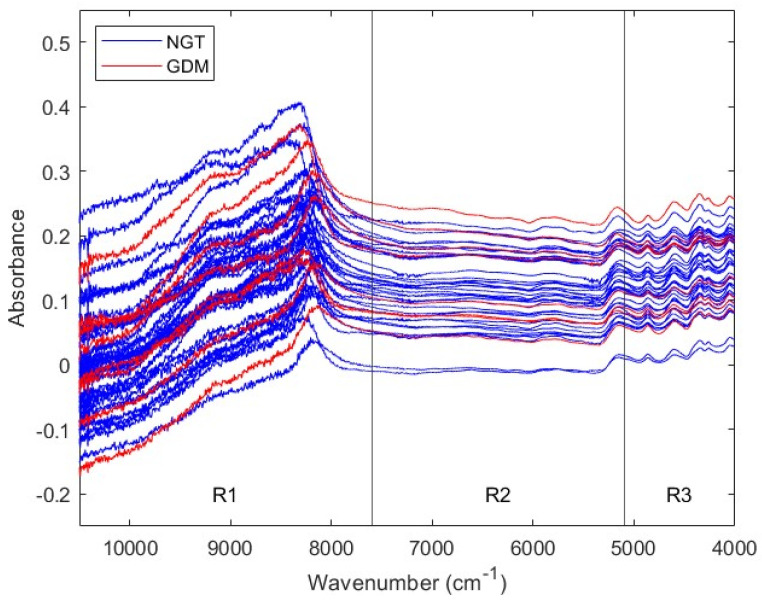
NIR spectra from second trimester serum samples. Each trace corresponds to the NIR spectrum of one serum sample, that is, of one subject. Spectra from NGT and GDM pregnant women are colored in blue and red, respectively. Four spectral ranges are considered: Full, from 10,500 to 4000 cm^−1^; Range 1, from 10,500 to 7600 cm^−1^; Range 2, from 7600 to 5100 cm^−1^; and Range 3, from 5100 to 4000 cm^−1^. NIR: near-infrared; NGT: normal glucose tolerance; GDM: gestational diabetes mellitus; R1: Range 1; R2: Range 2; R3: Range 3.

**Figure 4 biomedicines-12-01142-f004:**
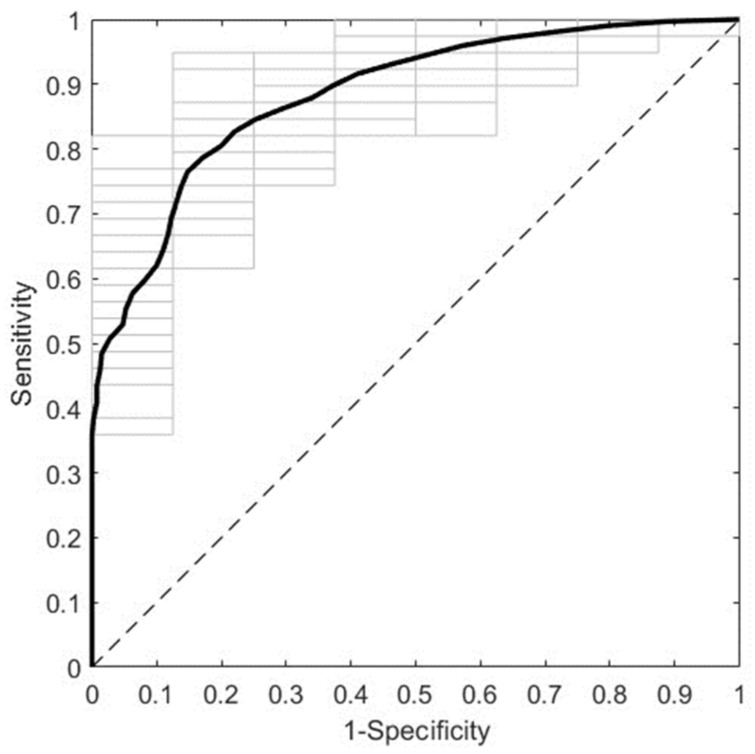
ROC curve of the best predictive model for the second trimester cohort. The model was trained with NIR spectra (R3, 5100–4000 cm^−1^) from second trimester serum samples after pretreatment by first derivative (width = 15) and mean centering. The average and the individual curves of 50 DCV repetitions are colored in black and gray, respectively. ROC: receiver operating characteristic; NIR: near-infrared; R3: Range 3; DCV: double cross-validation.

**Table 1 biomedicines-12-01142-t001:** Medical variables in the first trimester cohort.

Variable	Unit	NGT (*n* = 67)	GDM (*n* = 15)	*p* Value		All (*n* = 82)
Age	Years	30 ± 5	32 ± 7	0.394	NS	31 ± 6
BMI	Kg/m^2^	27.6 (23.3–31.2)	29.7 (26.6–31.6)	0.051	NS	28.0 (24.1–31.5)
Supplement consumption	%	64.2 (43/67)	53.3 (8/15)	0.557	NS	62.2 (51/82)
Hyperemesis	%	26.9 (18/67)	26.7 (4/15)	>0.999	NS	26.8 (22/82)
Vaginal bleeding	%	9.0 (6/67)	13.3 (2/15)	0.634	NS	9.8 (8/82)
Drug use before pregnancy	%					
Cigarettes		34.3 (23/67)	53.3 (8/15)	0.239	NS	37.8 (31/82)
Alcohol		53.7 (36/67)	60.0 (9/15)	0.777	NS	54.9 (45/82)
Other drugs		13.4 (9/67)	13.3 (2/15)	>0.999	NS	13.4 (11/82)
Prior pregnancy issues	%					
GDM		1.5 (1/67)	33.3 (5/15)	˂0.001	***	7.3 (6/82)
Hypertensive disorder		4.5 (3/67)	6.7 (1/15)	0.562	NS	4.9 (4/82)
Preterm birth		4.5 (3/67)	6.7 (1/15)	0.562	NS	4.9 (4/82)
Other		10.4 (7/67)	6.7 (1/15)	>0.999	NS	9.8 (8/82)
Prior non-viable pregnancy	%	20.9 (14/67)	20.0 (3/15)	>0.999	NS	20.7 (17/82)
Fertility problems	%	14.9 (10/67)	6.7 (1/15)	0.679	NS	13.4 (11/82)
PCOS	%	25.4 (17/67)	13.3 (2/15)	0.501	NS	23.2 (19/82)
First period age	Years	13 (12–14)	12 (11–13)	0.078	NS	13 (12–13)
Last period month	%			0.202	NS	
January		7.5 (5/67)	6.7 (1/15)			7.3 (6/82)
February		6.0 (4/67)	20.0 (3/15)			8.5 (7/82)
March		7.5 (5/67)	0.0 (0/15)			6.1 (5/82)
April		3.0 (2/67)	6.7 (1/15)			3.7 (3/82)
May		13.4 (9/67)	20.0 (3/15)			14.6 (12/82)
June		10.4 (7/67)	13.3 (2/15)			11.0 (9/82)
July		9.0 (6/67)	13.3 (2/15)			9.8 (8/82)
August		7.5 (5/67)	0.0 (0/15)			6.1 (5/82)
September		6.0 (4/67)	0.0 (0/15)			4.9 (4/82)
October		13.4 (9/67)	6.7 (1/15)			12.2 (10/82)
November		10.4 (7/67)	13.3 (2/15)			11.0 (9/82)
December		6.0 (4/67)	0.0 (0/15)			4.9 (4/82)
Personal morbid history	%					
Insulin resistance		3.0 (2/67)	6.7 (1/15)	0.459	NS	3.7 (3/82)
Thyroid dysfunction		4.5 (3/67)	6.7 (1/15)	0.562	NS	4.9 (4/82)
Asthma		6.0 (4/67)	0.0 (0/15)	>0.999	NS	4.9 (4/82)
Other		10.4 (7/67)	20.0 (3/15)	0.380	NS	12.2 (10/82)
Family morbid history	%					
Insulin resistance or prediabetes		3.0 (2/67)	6.7 (1/15)	0.459	NS	3.7 (3/82)
DM		32.8 (22/67)	66.7 (10/15)	0.020	*	39.0 (32/82)
Hypertension		41.8 (28/67)	60.0 (9/15)	0.255	NS	45.1 (37/82)
Hypothyroidism		17.9 (12/67)	33.3 (5/15)	0.287	NS	20.7 (17/82)
Hyperthyroidism		1.5 (1/67)	13.3 (2/15)	0.085	NS	3.7 (3/82)
Asthma		7.5 (5/67)	0.0 (0/15)	0.579	NS	6.1 (5/82)
Other		16.4 (11/67)	13.3 (2/15)	>0.999	NS	15.9 (13/82)

NGT: normal glucose tolerance; GDM: gestational diabetes mellitus; BMI: body mass index; PCOS: polycystic ovary syndrome; DM: diabetes mellitus; *: *p* < 0.05; ***: *p* < 0.001; NS: not significant.

**Table 2 biomedicines-12-01142-t002:** Predictive performance of the best ML models using NIR spectral data from first trimester serum samples.

Range ^a^	Pretreatment	Sp		Se		NER	
		Av	StD	Av	StD	Av	StD
Full	SM (W = 23) + N + MC	0.6946	0.0456	0.4507	0.0681	0.5726	0.0410
R1	N + MC	0.6722	0.0361	0.5920	0.0910	0.6321	0.0489
R2	SM (W = 3) + N + MC	0.5678	0.0322	0.6480	0.1035	0.6079	0.0542
R3	SM (W = 23) + MC	0.5931	0.0346	0.5133	0.0811	0.5532	0.0441

^a^ Full: 10,500–4000 cm^−1^; R1: 10,500–7600 cm^−1^; R2: 7600–5100 cm^−1^; R3: 5100–4000 cm^−1^. Sp: specificity; Se: sensitivity; NER: non-error rate; Av: average; StD: standard deviation; R1: Range 1; R2: Range 2; R3: Range 3; SM: smoothing; W: width; N: normalization; MC: mean centering.

**Table 3 biomedicines-12-01142-t003:** Medical variables in the second trimester cohort.

Variable	Unit	NGT (*n* = 39)	GDM (*n* = 8)	*p* Value		All (*n* = 47)
Age	Years	29 ± 5	30 ± 7	0.606	NS	29 ± 5
BMI	Kg/m^2^	27.0 ± 4.7	31.3 ± 6.5	0.034	*	27.7 ± 5.2
Supplement consumption	%	64.1 (25/39)	62.5 (5/8)	>0.999	NS	63.8 (30/47)
Hyperemesis	%	33.3 (13/39)	25.0 (2/8)	>0.999	NS	31.9 (15/47)
Vaginal bleeding	%	5.1 (2/39)	25.0 (2/8)	0.129	NS	8.5 (4/47)
Drug use before pregnancy	%					
Cigarettes		33.3 (13/39)	37.5 (3/8)	>0.999	NS	34.0 (16/47)
Alcohol		61.5 (24/39)	50.0 (4/8)	0.697	NS	59.6 (28/47)
Other drugs		25.6 (10/39)	0.0 (0/8)	0.174	NS	21.3 (10/47)
Prior pregnancy issues	%					
GDM		0.0 (0/39)	37.5 (3/8)	0.004	**	6.4 (3/47)
Hypertensive disorder		7.7 (3/39)	0.0 (0/8)	>0.999	NS	6.4 (3/47)
Preterm birth		5.1 (2/39)	12.5 (1/8)	0.436	NS	6.4 (3/47)
Other		7.7 (3/39)	0.0 (0/8)	>0.999	NS	6.4 (3/47)
Prior non-viable pregnancy	%	17.9 (7/39)	12.5 (1/8)	>0.999	NS	17.0 (8/47)
Fertility problems	%	17.9 (7/39)	0.0 (0/8)	0.329	NS	14.9 (7/47)
PCOS	%	25.6 (10/39)	12.5 (1/8)	0.659	NS	23.4 (11/47)
First period age	Years	13 (12–14)	12 (12–13)	0.058	NS	13 (12–14)
Last period month	%			0.729	NS	
January		2.6 (1/39)	0.0 (0/8)			2.1 (1/47)
February		7.7 (3/39)	25.0 (2/8)			10.6 (5/47)
March		12.8 (5/39)	0.0 (0/8)			10.6 (5/47)
April		5.1 (2/39)	12.5 (1/8)			6.4 (3/47)
May		12.8 (5/39)	0.0 (0/8)			10.6 (5/47)
June		10.3 (4/39)	12.5 (1/8)			10.6 (5/47)
July		15.4 (6/39)	25.0 (2/8)			17.0 (8/47)
August		12.8 (5/39)	0.0 (0/8)			10.6 (5/47)
September		2.6 (1/39)	12.5 (1/8)			4.3 (2/47)
October		10.3 (4/39)	12.5 (1/8)			10.6 (5/47)
November		5.1 (2/39)	0.0 (0/8)			4.3 (2/47)
December		2.6 (1/39)	0.0 (0/8)			2.1 (1/47)
Personal morbid history	%					
Insulin resistance		5.1 (2/39)	0.0 (0/8)	>0.999	NS	4.3 (2/47)
Thyroid dysfunction		10.3 (4/39)	0.0 (0/8)	>0.999	NS	8.5 (4/47)
Asthma		7.7 (3/39)	0.0 (0/8)	>0.999	NS	6.4 (3/47)
Other		10.3 (4/39)	37.5 (3/8)	0.084	NS	14.9 (7/47)
Family morbid history	%					
Insulin resistance or prediabetes		7.7 (3/39)	12.5 (1/8)	0.539	NS	8.5 (4/47)
DM		35.9 (14/39)	62.5 (5/8)	0.240	NS	40.4 (19/47)
Hypertension		48.7 (19/39)	62.5 (5/8)	0.701	NS	51.1 (24/47)
Hypothyroidism		17.9 (7/39)	25.0 (2/8)	0.639	NS	19.1 (9/47)
Hyperthyroidism		5.1 (2/39)	12.5 (1/8)	0.436	NS	6.4 (3/47)
Asthma		10.3 (4/39)	0.0 (0/8)	>0.999	NS	8.5 (4/47)
Other		12.8 (5/39)	12.5 (1/8)	>0.999	NS	12.8 (6/47)

NGT: normal glucose tolerance; GDM: gestational diabetes mellitus; BMI: body mass index; PCOS: polycystic ovary syndrome; DM: diabetes mellitus; *: *p* < 0.05; **: *p* < 0.01; NS: not significant.

**Table 4 biomedicines-12-01142-t004:** Predictive performance of the best ML models using NIR spectral data from second trimester serum samples.

Range ^a^	Pretreatment	Sp		Se		NER	
		Av	StD	Av	StD	Av	StD
Full	2D (W = 15) + N + MC	0.8133	0.0324	0.1150	0.0556	0.4642	0.0321
R1	WLS + N + MC	0.8754	0.0414	0.1625	0.1218	0.5189	0.0643
R2	2D (W = 3) + N + MC	0.6821	0.0288	0.3875	0.1191	0.5348	0.0613
R3	1D (W = 15) + MC	0.8713	0.0361	0.7075	0.0783	0.7894	0.0431

^a^ Full: 10,500–4000 cm^−1^; R1: 10,500–7600 cm^−1^; R2: 7600–5100 cm^−1^; R3: 5100–4000 cm^−1^. Sp: specificity; Se: sensitivity; NER: non-error rate; Av: average; StD: standard deviation; R1: Range 1; R2: Range 2; R3: Range 3; 2D: second derivative; W: width; N: normalization; MC: mean centering; WLS: weighted least squares; 1D: first derivative.

**Table 5 biomedicines-12-01142-t005:** Comparison of the method developed in this study for the diagnostic support of GDM with those presented in similar articles.

Time of Application	Study	Instrumental Technique	Predictive Power	Duration of Analysis
Before diagnosis of GDM by OGTT	This study	NIRS	AUROC: 0.5768 ± 0.0635 NER: 0.6321 ± 0.0489	32 min
	[21]	LC-MS	AUROC: 0.724–0.902	>8 h
	[48]	LC-MS	AUROC: 0.7075	1.5 h
	[49]	LC-MS	AUROC: 0.729–0.906	>4 h
	[22]	GC-MS	AUROC: 0.771–0.907	>1.5 h
	[50]	GC-MS	AUROC: 0.745–0.797	>16 h
	[23]	NMRS	AUROC: 0.796	Not mentioned. Typically 1–1.5 h [51]
	[52]	NMRS	AUROC: 0.59	
	[53]	NMRS	NER: 0.635–0.825	
	[54]	NMRS	AUROC: 0.610–0.719	
	[27]	PCR	AUROC: 0.7694	>8 h
	[55]	PCR	NER: 0.531–0.552	>2 h
	[56]	PCR	AUROC: 0.600–0.669	>2.5 h
At the time of diagnosis of GDM by OGTT	This study	NIRS	AUROC: 0.8836 ± 0.0259 NER: 0.7894 ± 0.0431	32 min
	[48]	LC-MS	AUROC: 0.7800	1.5 h
	[50]	GC-MS	AUROC: 0.745–0.828	>16 h
	[57]	GC-MS	AUROC: 0.83–0.90	>16 h
	[52]	NMRS	AUROC: 0.62	Not mentioned. Typically 1–1.5 h [51]
	[53]	NMRS	NER: 0.695–0.885	
	[27]	PCR	AUROC: 0.7694	>8 h
	[55]	PCR	NER: 0.531–0.552	>2 h
	[58]	PCR	AUROC: 0.74–0.92	>2.5 h

GDM: gestational diabetes mellitus; OGTT: oral glucose tolerance test; NIRS: near-infrared spectroscopy; LC-MS: liquid chromatography coupled to mass spectrometry; GC-MS: gas chromatography coupled to mass spectrometry; NMRS: nuclear magnetic resonance spectroscopy; PCR: polymerase chain reaction; AUROC: area under the receiver operating characteristic curve; NER: non-error rate.

## Data Availability

The original contributions presented in the study are included in the article/Appendix A, further inquiries can be directed to the corresponding author/s.

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
