# Peer review of "Simple and Fast Prediction of Gestational Diabetes Mellitus Based on Machine Learning and Near-Infrared Spectra of Serum: A Proof of Concept Study at Different Stages of Pregnancy"

_biomedicines, 2024, doi:10.3390/biomedicines12061142_

Round 1

Reviewer 1 Report

Comments and Suggestions for Authors

In my reading, the title needs improvement, as it should be clear that the NIRS analysis has been done on blood samples. Furthermore, it should be clearer that the study is exploratory in its nature.

Assuming that the OGT is the golden standard, the overall performance at OGT testing is not that impressive. Can you convert this to sensitivity and specificity compared to the golden standard ? You have also collected 28 medical variables. It should therefore be possible to compare the performance in the first trimester of pregnancy with some of these characteristics like BMI (not just addition).

Were only first pregnancies included, or also multipara (if so, history of GDM accepted, tables in the results suggest otherwise ?)

The batch analysis is well understood for the study, but likely not useful in a potential future clinical application (simple and fast ? ). Or should I more understand the reported findings to reflect and unveil mechanisms of disease ? However, a ML approach likely makes it difficult to explore.

Is it correct that women were included in either cohort 1, or cohort 2 (so unpaired).

If I understand the stats description well, there was no a priori hypothesis or power calculation, so that the wording on causality should be more cautious.

Figure 2 does suggest that the overall performance is still quite poor in the first trimester, while the figure 4 should be weighted on performance versus the OGT results.

Reviewer 2 Report

Comments and Suggestions for Authors

The authors developed two machine learning models for predicting gestational DM at different stages of pregnancy. These models aimed to provide a simple and fast tool for medical diagnosis. I suggest the authors address the following comments to enhance the paper's quality.
Comment 1. Abstract: State the percentage range improvement by the proposed work in terms of ROC and time compared with the existing works.
Comment 2. More terms can be included in the keywords to reflect the scope of the paper better.
Comment 3. Section 1 Introduction:
(a) Provide the prevalence and severity of gestational DM at different stages of pregnancy.
(b) Update the content with the latest references. Focus mainly on recent 5-year articles.
(c) Add a literature review to summarize the methodologies, results, and limitations of the latest ML-based works for gestational DM.
(d) Clearly state the research contributions, preferably in point form.
Comment 4. Section 2:
(a) Add an introductory paragraph before Subsection 2.1.
(b) Subsection 2.2, how many participants were involved in the data collection?
(c) Were there any experts who could verify the data quality?
(d) Subsection 2.8 ML Analyses:
- This subsection should be the methodology of ML approaches. Therefore, please provide justification for selecting these approaches. Share the approaches' designs and formulations, including equations, pseudo-code, etc. Discuss how you optimally design the models.
Comment 5. Section 3:
(a) Add an introductory paragraph before Subsection 3.1.
(b) Table 1: what is “NS”?
(c) Figure 1: Why are there multiple curves in each of NGT and GDM? How can we distinguish between individual curves?
(d) Table 2: What is the model number? How many numbers are there?
Comment 6. A comparison between the proposed work and the existing works is expected.

Comments on the Quality of English Language

Correct some typos and minor grammatical mistakes.

Round 2

Reviewer 1 Report

Comments and Suggestions for Authors

i value the efforts made to further adapt and revised the paper, to improve the reporting, and to adapt the conclusions and impact reflections. I suggest to accept. 

Reviewer 2 Report

Comments and Suggestions for Authors

There are several follow-up comments to be considered by the authors:
Follow-up comment 1: The abstract should clearly state the percentage improvement by the proposed work compared with the existing works.
Follow-up comment 2: More terms should be included in the keywords to better reflect the scope of the paper. In addition, “prediction” and “screening” are not precise terms.
Follow-up comment 3: Avoid citing many references in a single sentence without further elaboration. Please refer to “GDM pregnancies [12–17]”, “GDM diagnosis. [21–29]”, etc.
Follow-up comment 4: The works discussed in the literature review were considered relevant papers for comparison in Section 4. In addition, please provide a clear comparison between the authors’ work and the existing works in a table format.
Follow-up comment 5: It causes confusion that Section 4 also covers content related to the results.
